# Combining Oncolytic Viruses and Small Molecule Therapeutics: Mutual Benefits

**DOI:** 10.3390/cancers13143386

**Published:** 2021-07-06

**Authors:** Bart Spiesschaert, Katharina Angerer, John Park, Guido Wollmann

**Affiliations:** 1Christian Doppler Laboratory for Viral Immunotherapy of Cancer, Medical University Innsbruck, 6020 Innsbruck, Austria; bart.spiesschaert@boehringer-ingelheim.com (B.S.); Katharina.Angerer@i-med.ac.at (K.A.); 2Institute of Virology, Medical University Innsbruck, 6020 Innsbruck, Austria; 3ViraTherapeutics GmbH, 6063 Rum, Austria; 4Boehringer Ingelheim Pharma GmbH & Co. KG, 88397 Biberach a.d. Riss, Germany; john.park@boehringer-ingelheim.com

**Keywords:** oncolytic virus, small molecule, cancer immune therapy, combination therapy, cancer therapy, immunotherapy

## Abstract

**Simple Summary:**

Oncolytic viruses can be a potent tool in the fight against cancer. However, in clinical settings their ability to replicate in and kill tumors is often limited. Combinations with specific small molecule compounds can address some of these limitations and help oncolytic viruses reach their full potential. The aim of this review is to provide an overview of the different types of small molecules with which oncolytic viruses can achieve therapeutic synergy. We focus on the underlying mechanisms in three functional areas: combinations that increase viral replication, enhance tumor cell killing and improve antitumor immune responses.

**Abstract:**

The focus of treating cancer with oncolytic viruses (OVs) has increasingly shifted towards achieving efficacy through the induction and augmentation of an antitumor immune response. However, innate antiviral responses can limit the activity of many OVs within the tumor and several immunosuppressive factors can hamper any subsequent antitumor immune responses. In recent decades, numerous small molecule compounds that either inhibit the immunosuppressive features of tumor cells or antagonize antiviral immunity have been developed and tested for. Here we comprehensively review small molecule compounds that can achieve therapeutic synergy with OVs. We also elaborate on the mechanisms by which these treatments elicit anti-tumor effects as monotherapies and how these complement OV treatment.

## 1. Introduction

In the course of oncogenic transformation and progression, tumor cells acquire distinct features that have been termed hallmarks of cancer [1,2]. Some of these aberrations form the base for the tumor-preferential infection and propagation of natural or recombinant oncolytic viruses (OVs) [3]. Evasion of growth suppressive mechanisms, continuous proliferative signaling, unrestricted replication machinery and the evasion of innate and adaptive immune control constitute characteristics that can be exploited by OVs. In general, naturally occurring or genetically engineered virotherapy candidate viruses share the core features of tumor-preferential infection, replication, and lysis. Beyond that, they display the diversity of viruses on multiple levels: human pathogen-derived versus animal viruses, DNA versus RNA genome, enveloped versus non-enveloped, nuclear versus cytosolic replication cycle, etc. [4]. Herpes simplex virus (HSV) and adenovirus (AdV) are human pathogenic DNA viruses that have been developed for three decades as oncolytic agents with a plethora of modified variants being tested in preclinical and clinical settings. This resulted in the first regulatory approvals of H101, a genetically engineered adenovirus, in 2005 in China and talimogene laherparepvec (T-VEC), a recombinant attenuated HSV-1 with a transgene encoding for granulocyte-macrophage colony-stimulating factor (GM-CSF), in 2015 in the USA and Europe [5]. Development of oncolytic HSV and AdV variants has continued though with a strong focus on next generation “armed” OVs expressing a multitude of immune modulatory transgenes. Another clinically advanced oncolytic platform is based on the vaccinia virus (VV), a large DNA virus encoding about 200 genes with an exclusive cytosolic replication cycle. Its ability to accommodate up to 40 kb of transgene DNA make VV a prime platform for arming with immune modulatory cargo genes [6]. A related member of the poxvirus family, myxoma virus, has also extensively been explored as an oncolytic agent in pre-clinical settings [7]. H1, a small rat parvovirus, completes the list of the major DNA-based oncolytic agents. This natural onco-preference is in large part based on a dependency on proliferating cells and signaling pathway aberrations [8]. Reovirus, a natural occurring human virus with double stranded RNA genome, is usually not associated with disease in adults and its onco-tropism was originally thought to be linked to RAS transformation in cancer cells, although recent data suggest a more multifactorial relationship [9]. The Edmonston vaccine strain of measles virus, a negative strand RNA paramyxovirus, displays a certain natural onco-selectivity in part due to frequent overexpression of its receptor, CD46, in a range of different cancer types [10]. Newcastle disease virus, an avian paramyxovirus without causing known human disease, harbors a natural onco-selectivity due to interaction with anti-apoptotic proteins and its dependence on a defective antiviral make-up frequently observed in cancer cells [11]. Vesicular stomatitis virus (VSV), a negative strand RNA virus of the rhabdoviridae family, causes mild disease in livestock with clinical symptoms rarely reported in humans. Its ubiquitous receptor entry translates to a pan-tropism for a very broad range of tumor types, but also holds the potential for some neuro-toxicity once it can access the brain. Consequently, VSV development has long been driven by attenuation strategies [12]. As with several other RNA viruses, the primary mode of onco-selectivity is based on reduced antiviral defense mechanisms in certain tumors [13]. In recent years, a large number of VSV variants armed with immunomodulatory transgenes has been tested in preclinical settings and in early phase clinical testing [14]. With few exceptions, most OVs are rather sensitive to innate antiviral control. This increases their safety aspect towards normal cells while letting them take advantage of impaired innate immune signaling in tumors [13]. These OVs are therefore also considerably better suited to be combined with small molecules that counter innate antiviral immunity. During early OV developments, the paradigm was that the efficacy of OV treatment correlated to virus replication. Viral spread throughout the tumor, and subsequent OV-mediated cancer cell lysis, were thought to be the main drivers of OV therapy [15]. According to this thinking, OVs were initially combined with immunosuppressive small molecule compounds in order to limit the antiviral immune response and allow OVs to replicate to higher titers within the treated tumors [16,17]. The different mechanisms and compounds that modulate the innate antiviral immunity are discussed in detail below. Such approaches have yielded promising results mostly in preclinical settings [18]. However, the modes of action by which OVs can be therapeutic are more complex in immunocompetent patients and the immune activating potential of OVs has increasingly dominated the discussion [19,20,21]. OV treatments are now considered potent partners for immunotherapies [22]. Few treatment modalities inherently hold the potential to simultaneously induce immunogenic cell death (ICD), stimulate innate and adaptive immune responses, enhance T cell infiltration and repolarize an immune-suppressive tumor microenvironment (TME) [23,24,25]. Immunogenic cell death is associated with the induction and release of pro-inflammatory cytokines and danger-associated molecular patterns (DAMPs) [26]. DAMPs are especially expressed when infected cells die in an immunogenic manner, such as necroptosis. Enhancing these modes of cell death through the combination with tumor cell death enhancing (TCDE) small molecule compounds has therefore become a central focus [27,28] and is also discussed in detail below. The presence of virus related pathogen associated molecular patterns (PAMPs) and DAMPs subsequently facilitates the attraction of immune cells which contribute to the immune-stimulatory state by producing additional inflammatory cytokines [29]. This can eventually shift the immunosuppressive TME allowing a successful antitumor immune response to occur [30,31]. Still, even after induction of an antitumor immune response, the continuous reshaping of the TME at later stages constitutes further challenges [26]. For example, OV treatment commonly induces the expression of programmed cell death ligand 1 (PD-L1). However, this can be successfully countered by immune checkpoint inhibiting antibodies [32]. Small molecule checkpoint inhibitors could contribute to OV treatment in a similar fashion [33]. Other components of the TME, such as tumor growth factor (TGF)-β, epigenetic major histocompatibility complex (MHC) repression, cytotoxic T-lymphocyte-associated Protein 4 (CTLA-4), T-cell immunoglobulin and mucin-domain containing-3 (TIM-3), etc.), regulatory T-cells (Treg), myeloid-derived suppressor cells (MDSC), and M2 tumor associated macrophages (TAMs) can also contribute to an immunosuppressive therapy-resistant state. Some of these factors can be targeted by small molecule therapeutics [34], which will also be discussed in a separate section below. As we show in the following, the different aspects of multimodal OV treatment can be improved by a vast array of small molecule compounds, and a future impact on improving the clinical outcome of such combinations is conceivable.

## 2. Combinations Affecting Viral Propagation in Tumor Cells

The selectivity of various oncolytic viruses largely depends on defects in the tumor cell’s innate ability to fend off viral infections [35]. However, the initial assumption that an impaired interferon (IFN) response is a common feature shared by many tumors [36] may not reflect the clinical reality of solid cancers’ heterogeneity [37]. Some tumors, such as pancreas cancer, may even display an upregulated antiviral state leading to primary resistance [38]. A constitutive interferon pathway activation was also described as a main determinant for oncolytic measles virus activity in a human glioblastoma specimen [39]. On the other hand, tumors induced by oncoviruses, such as HPV-associated cervical or head and neck cancers, tend to frequently display strongly impaired antiviral innate responses [40]. However, in light of missing systematic assessments of a large range of tumor types, general conclusions as to what cancer types are more antivirally active and which are not remain to be drawn. Although most viruses have evolved to express proteins that counter antiviral measures [41], engineering of many oncolytic viruses were aimed at abolishing exactly those viral counter measures, generating OVs with a heightened IFN sensitivity [37]. Cornerstones of the antiviral innate immune response are type I (and to a lesser extend type III) interferons [42]. Both IFN types converge in their signaling and induce transcriptional responses through the Janus kinase signal transducers and activators of transcription (JAK/STAT) pathway [43]. Their signaling is associated with downstream expression of interferon stimulated genes (ISGs) which act as antiviral effector proteins countering viral replication. OV replication is impaired when these pathways are still intact in the treated tumor cells [44]. In the following, we will discuss various compound classes involved in inhibiting antiviral signaling pathways and which hold the potential to either enhance replication of OVs or even address OV resistance in cancer cells.

### 2.1. JAK-STAT Signaling Inhibition

Inhibitors of Janus kinases (JAK), such as JAK inhibitor I (a pan-JAK inhibitor) or ruxolitinib (a specific JAK1/2 inhibitor) (Figure 1), were able to rescue the replication of VSV in several human pancreatic ductal adenocarcinoma (PDA) cells that were otherwise resistant due to constitutive high-level expression of certain interferon stimulated genes (ISGs) [38,45,46]. This effect was improved even further when Polybrene or DEAE-dextran were additionally added, improving VSV attachment and entry and allowing more cells to be infected [47]. A similar effect was seen for refractory human head and neck squamous cell carcinoma (HNSCC) cell lines which owed their VSV resistance to the constitutive expression of a different set of ISGs. Here, JAK inhibitor I and ruxolitinib were also successful in rescuing virus replication with a 100- to 1000-fold increase in yield. Interestingly, other innate immune small molecule compounds, such as histone deacetylase inhibitors (HDI; LBH589), phosphoinositide 3-kinase (PI3K) inhibitors (GDC-0941, LY294002), mammalian target of rapamycin complex 1 (mTORC1) inhibitors (rapamycin) or STAT3 inhibitor VII were not effective [48]. Combination therapy with ruxolitinib and VSV-IFNβ also enhanced viral replication and oncolysis in several non-small cell lung cancer (NSCLC) cell lines [49]. However, several of these compounds were effective in rescuing OV replication in other tumor cell types, as discussed in the sections below underlining the heterogeneity in mechanisms among different tumor cells by which synergy with OVs can occur. In melanoma, mutations in the JAK1/2 signaling pathway as well as JAK1/2 inhibition increase sensitivity to VSV-dM51 [50]. The dual inhibitor of JAK1 and IκB kinase (IKK), TPCA-1 was also shown to improve HSV replication of malignant peripheral nerve sheath tumor (MPNST) cells [51]. OVs that replicate in the cytoplasm, such as RNA viruses and poxviruses, can also trigger direct antiviral effector responses that can hamper their replication and subsequent oncolytic effects. Viral RNA activates the cytosolic PKR by inducing dimerization and subsequent auto-phosphorylation reactions. The protein kinase R (PKR) pathway leads to a stress response by activating other pathways such as the interconnected nuclear factor κ-light chain enhancer of activated B cells (NF-κB) & c-Jun N-terminal kinase (JNK) pathways (Figure 2) [52,53,54]. JNK are kinases involved in a diverse set of cellular functions, ranging from cell death, survival and proliferation to innate immunity [54]. Specifically, JNK are essential for the expression regulation of many immune mediator genes, such as cytokines (e.g., interleukins (ILs) IL-2, IL-4, IL-8, IL-18, IFN-γ, granulocyte-macrophage colony-stimulating factor (GM-CSF), C-C motif chemokine ligand 5 (CCL5), tumor necrosis factor α (TNF-α)) [55,56,57,58,59] and adhesion molecules (ICAM-1) [53]. While JNK inhibition has been reported to act antivirally on encephalomyocarditis virus, rotavirus and HSV [60,61,62], a virus promoting effect was seen for vaccinia virus. Here, murine embryonic fibroblasts devoid of JNK showed a significant increase in titer. In line with these results, an increase of apoptosis was seen when wildtype murine embryonic fibroblast cells were co-treated with the JNK-specific inhibitor SP600125 [55,63]. This suggests that JNK inhibition, at least under very specific conditions, can be beneficial for OV therapy [63].

### 2.2. Inhibition of NF-kB Signaling

Nuclear factor (NF)-κB and inhibitor of NF-κB kinase (IKK) proteins regulate many cellular responses to stimuli, such as innate and adaptive immunity, cell death, and inflammation [64]. NF-κB and IKK therefore play key roles in regulating the innate immune response against OVs. Indeed, two types of compounds enhance OV replication through very distinct mechanisms at different stages of NF-κB-mediated transcription [65]. For instance, fumaric and maleic acid esters, such as dimethyl fumarate (DMF), block the nuclear translocation of NF-κB and have been shown to improve replication of several OVs and subsequent therapeutic outcomes by inhibiting type I IFN [66]. Another point of intervention is in the nucleus after NF-κB has already bound DNA [67]. At this point triptolide blocks transcription, leading to an increase of VSV replication in several VSV-resistant tumor cell types (Figure 2) [68]. Before NF-κB can facilitate transcription of innate immune genes it has to be released from the IκB kinase β (IKKβ) complex. The activation of IKKβ, by the phosphorylation of IκBα and its subsequent proteasomal degradation, allows NF-κB to relocate to the nucleus [64]. Blocking IKKβ can be therapeutically exploited since NF-κB is overexpressed in many cancer types [69]. Consequently, inhibiting IKKβ shows much promise for synergizing with OVs (Figure 2). This would be especially advantageous for OVs, such as VSV and NDV, that rely on defective innate immunity for their onco-selectivity [70]. This was confirmed in studies on malignant peripheral nerve sheath tumor cells and some pancreatic ductal adenocarcinoma cell lines that showed resistance to oncolytic HSV and VSV infection, respectively. In combination with the IKKβ inhibitor TPCA-1, this resistance was overcome and productive infection was achieved [46,51].

### 2.3. PI3K/AKT/mTOR Pathway Antagonists

Important for cell survival and growth, the phosphoinositide 3-kinase (PI3K)/Ak strain transforming (AKT)/mTOR pathway is also crucially involved in the induction of type 1 interferons (Figure 3) [71]. It is commonly activated in numerous types of cancer [72] via mutations or amplification of genes encoding receptor tyrosine kinases, subunits of PI3K, AKT or activating isoforms of rat sarcoma (Ras) [73]. The first agents, targeting the PI3K pathway with the specific purpose of treating cancer, were analogues of rapamycin, namely everolimus (RAD 001) and temsirolimus [73]. Hence, inhibition of mTOR is expected to augment the oncolytic activity particularly of those viruses depending on impaired antiviral responses within a tumor cell. The macrolide compound rapamycin is a prototypical inhibitor of the serine/threonine protein kinase mTOR. Combining rapamycin with the highly IFN-sensitive VSV-mutant strain (VSVΔM51) led to significant increase of the oncolytic effect [74]. In addition other oncolytic RNA viruses, such as NDV and reovirus, showed improved oncolytic effect in mice when co-treated with rapamycin [75,76]. Oncolytic DNA viruses also benefit from co-treatment with rapamycin. The yield and dissemination of an HSV-derived oncolytic virus was markedly increased in semi-permissive tumor cell lines [77]. An oncolytic vaccinia virus (VACV) only achieved complete remission in in vivo models when it was combined with rapamycin [78]. A key restriction factor for myxoma virus in human cells is its dependence on AKT activation [79]. By inhibiting mTORC1, AKT becomes hyperactivated through the release from the negative feedback loop between ribosomal protein S6 kinase beta-1 (S6K1) and insulin receptor substrate 1 (IRS-1) [80]. This subsequently enhances myxoma virus replication which also translates to increased survival in vivo [81,82,83]. mTOR inhibition can also lead to a decrease in phosphorylation of the effector proteins, eukaryotic translation initiation factor 4E-binding protein 1 (4E-BPs) and S6Ks, which are essential for type I interferon (IFN) production (Figure 3) [84,85]. This inhibition of the type I interferon response also contributes to a more pronounced replication of myxoma virus in vitro and increased efficacy in vivo [86]. Everolimus was tested in combination with an oncolytic adenovirus. Even though, in vitro, RAD001 seemed to interfere with the viral replication, potent anti-glioma effects were seen in vivo. This was presumably due to the induction of autophagic cell death [87,88]. Increased efficacy through modulation of autophagy in similar settings is also described for other OVs [75,76]. The hyperactivation of AKT during mTORC1 inhibition might have benefits when combined with myxoma virus [81,82,83], but in other settings can have a negative effect on survival. In phosphate and tensin homolog (PTEN)-deficient glioblastoma patients, for instance, hyperactivation of AKT, following rapamycin treatment, was associated with more rapid onset of tumor progression [89]. The mTORC2 complex, which is insensitive to rapamycin and its analogues, activates AKT and has a distinct role in tumor maintenance and progression [90]. For OVs with a dependency on a weakened antiviral state within the tumor, mTORC2 antagonists that also inhibit mTORC1 would be a superior option. ATP-competitive mTOR kinase inhibitors (TKIs) achieve this by targeting the kinase domain of mTOR, thereby also blocking the activation feedback of PI3K/Akt signaling (Figure 3) [91]. Indeed, mTORC1/2 inhibitors, such as PP242, INK1341, INK128 or Torin1, were also able to increase HSV replication and oncolysis by altering eIF4E/4E-BPs expression [77]. Specific inhibitors, such as rapamycin and TKIs, are prone to trigger the development of secondary resistance after prolonged treatment [92]. Consequently, inhibitors were developed that target the same signaling pathway but at multiple sites. Dual PI3K/mTOR inhibitors, such as voxtalisib [93], target the p110α, β, and γ isoforms of PI3K as well as the ATP-binding sites of both mTORC1 and mTORC2, completely suppressing PI3K/Akt signaling [91]. Combinations with OVs have yet to be reported. BKM120, another pan-class PI3K inhibitor, targeting all four catalytic isoforms, in combination with oncolytic HSV-1, was effective in the treatment of Du145 prostate cancer sphere forming cells (PCSCs) [94]. Finally, the benefits of combining PI3K/Akt signaling blockade and OVs can also work in the opposite direction, demonstrated by the combination of an oncolytic HSV and PI3K/Akt inhibitors (LY294002, triciribine, GDC-0941, BEZ235). Here, treatment with the OV sensitized the tumor cells to the inhibitors through enhanced Akt activation [95,96]. Indirectly, PI3K inhibitors, more specifically PI3Kδ-selective inhibitors, could improve systemic OV delivery to tumors through attachment inhibition of systemic macrophages [97].

### 2.4. Proteasome Inhibitors

Another approach to indirectly inhibit NF-κB is by blocking proteasomal degradation. The rationale is that proteasome inhibition blocks NF-kBs release from the IKKβ complex (Figure 2). Indeed, the proteasome inhibitor bortezomib improved the viral replication of oncolytic HSV and also enhanced necroptotic tumor cell death through increased endo-plasmatic reticulum (ER) stress and unfolded protein response (UPR) (Figure 4C) [98,99,100]. However, when bortezomib was combined with VSV, a reduction in replication and spread was seen in myeloma cells despite NF-κB activation being blocked. Interestingly, despite these antagonistic effects in vitro, co-treatment in vivo did improve the antitumor efficacy [101]. Similarly, another proteasome inhibitor PS-341 blocked the replication of VSV in human adenocarcinoma A549 cells [102] and infection with HSV strains. These seemingly contradictory studies make the combination of proteasome inhibitors and OVs a treatment option that needs to be further elucidated.

### 2.5. Tankyrase Inhibition

Resistance to PI3/AktT inhibitors is linked to Wnt/b-catenin signaling hyperactivation [103] and can be countered by the Wnt/tankyrase inhibitor NVP-TNKS656 [104]. Hence a direct synergy between tankyrase inhibitors (TNKSi) and OVs might be possible. Tankyrases play a role in the replication of different herpes viruses. The inhibition of tankyrase has been shown to promote replication of beta- (cytomegalovirus) and gamma-herpesvirus (Epstein-Barr virus), with the underlying mechanism via which this benefits the virus still to be elucidated [105,106]. In contrast, TNKS inhibition acts suppressive on the alpha-herpesvirus, HSV-1 [104]. However, direct combination regimens of TNKSi and OVs have not yet been published, but such studies might be merited.

### 2.6. Receptor Tyrosine Kinase Inhibitor

In the antiviral context, direct inhibition of PKR and Rnase was also achieved by another class of small molecule compounds. The ATP-competitive inhibitor of vascular endothelial growth factor (VEGF) and platelet derived growth factor (PDGF) receptors, sunitinib, was reported also to be a strong inhibitor for both PKR and RnaseL [107] (Figure 2). These compounds also have more direct impact on tumor growth through their negative regulation of tumor vascularization. Due to their broader mode of action this group of inhibitors can be referred to in more general terms as receptor tyrosine kinase inhibitors (RTKIs). These compounds proved to be very beneficial when combined with oncolytic VSV, leading to the elimination of prostate, breast, and kidney malignant tumors in mice [108]. Synergistic effects with RTKIs were also shown for vaccinia and reovirus in pancreatic neuroendocrine tumors and renal cell carcinoma, respectively [109,110], as well as for the combination with HSV in glioblastoma [111]. However, vaccine virus is also connected to the activation of the epidermal growth factor receptor (EGFR) pathway for their replication and spread. Here, simultaneous administration of RTKIs, such as imatinib and sorafenib, resulted in the inhibition of vaccinia virus replication [112,113]. Nonetheless, oncolytic vaccinia virotherapy, followed by sorafenib treatment, showed enhanced efficacy compared to either monotherapy. This is most likely due to OV-mediated sensitization of the tumor cells and tumor vasculature to VEGF/VEGFR inhibitors [112]. Part of these reported benefits are also achieved through modulation of the tumor microenvironment. When MC38 tumor bearing mice were pretreated with sunitinib, the anti-tumor response, induced by a tumor associated antigen (TAA)-armed virus, was markedly improved through a decrease in inhibitory regulatory T cells (Tregs) and myeloid-derived suppressor cells (MDSCs) after sunitinib treatment [114]. This adaptive immune modulation is achieved by interacting with RTKs expressed on regulatory immune cell populations, such as c-KIT and VEGFR-1 [115,116]. In a similar setting, the more broad-range RTK inhibitor cabozantinib also showed a more diverse and potent effect and immunomodulatory effects with additional expression of MHC-I molecules, ICAM-1, Fas, and calreticulin on tumor cells. Modulation of antigen expression is most likely to be facilitated by its hepatocyte growth factor receptor (MET) inhibition [117]. Another more specific EGFR inhibitor, erlotinib, also seems to enhance the oncolytic effect in some human pancreatic cancer cells through a similar mechanism for oncolytic HSV. Here, prolonged viral presence was reported [118]. On the other hand, in tumors, characterized by upregulated EGFR signaling, the synergism seemed predominantly driven by a concerted antiangiogenic effect [119].

### 2.7. Histone Deacetylase Inhibitors (HDIs)

Transcription regulation requires deacetylase activity [120]. Histone deacetylase inhibitor compounds (HDIs) were found to rescue viral replication in resistant cells [120,121,122], which led to several investigations into the potential to augment OV replication. Interestingly, the blunting of the antiviral response (Figure 1) seemed to be limited to tumor cells, leaving the inhibition of viral replication in normal tissue intact [17]. However, an enhanced effect was also seen in proliferating endothelial cells [123]. The mechanism by which this specificity occurs remains unclear. It is suggested that this might be due to either an inherent preference of OVs for tumor cells or an enhanced susceptibility of tumor cells for these small molecules [124]. This enhanced susceptibility could be caused by the aberrant activity of histone deacetylases (HDACs), documented for several types of cancers [125,126,127]. Numerous HDI/OV combinations were tested in different tumor models showing the therapeutic benefit of blunting the innate antiviral response during OV treatment (Table 1). Some HDIs, such as butyrate and trichostatin A (TSA), can also indirectly inhibit the innate immune signaling through the inhibition of NF-κB activation by reducing proteasome subunit expression [128]. Apart from inhibiting the innate immune response, the adaptive immune response was also beneficially influenced with entinostat resulting in prolonged lymphopenia and depletion of Tregs [129,130,131]. Another HDI, valproate, was shown to suppress production of IFN-γ, and immune cell infiltration including NK cells, macrophages and lymphocytes, which helped promote virus growth but also has the potential to dampen anti-tumor immune responses [130,132,133,134]. This discrepancy in modulating the adaptive immune response can be related to the differences in HDAC targets of the different HDIs. Trichostatin A inhibits class I and II HDACS [135], Entinostat inhibits class I HDACs [136], whereas vorinostat and to a lesser extent valproate are pan-HDAC inhibitors [137,138]. Among the HDIs vorinostat is considered the more potent candidate for combination with OVs. However, more recent screenings have uncovered an even more potent compound to promote viral replication in less permissive tumors, namely viral sensitizer 1 and analog 28 (VSe1-28). This increased viral yield of VSV up to 2000 fold in vitro [124]. Further, reovirus has recently been described to synergize with HDAC inhibitor belinostat in both sensitive and belinostat-resistant T cell lymphoma cells [139].

In addition, the HDI trichostatin has been reported to increase expression of MHC-I molecules on the cell surface [146]. This is of particular interest for OVs used in a cancer vaccine setting, where downregulation of MHC-I expression can result in a relapse [147]. This increased MHC-I expression was further improved when trichostatin was combined with the hypomethylation agent, 5-azacytidine [146,148]. Beyond the interference with the innate antiviral activity and stimulating effects on the adaptive immune responses, HDIs have also been shown to enhance the direct tumor cell killing and replication of H1 parvovirus by increasing the acetylation of the viral NS-1 protein [144].

## 3. Combinations Enhancing Tumor Cell Death

Evasion of cell death is one of the main hallmarks of cancer. Apoptosis resistance develops frequently by either upregulation of anti-apoptotic elements or countering pro-apoptotic stimuli [149]. Though less prominent, other forms of programmed cell death can be similarly overridden, such as necroptosis [150]. Of note, some viruses employ analogous strategies to counter cell death as an archetypal cellular defense mechanism against viral infection, exemplified by the oncolytic HSV [151] and vaccinia virus [152]. Consequently, viral oncolysis alone rarely leads to widespread and complete cell death, opening the door for a combination approach with cell death sensitizers. Another aspect of such combinations links the aforementioned often limited intra-tumoral spread of OVs with the potential of bystander killing of uninfected cells [153]. Sensitizing a tumor mass with agents promoting cell death has been shown to significantly increase the kill zone of oncolytic viruses beyond the infected areas, yet still confined to the tumor [154]. The following section gives an overview of small molecule compounds that augment tumor cell killing and thus hold promise to synergize with oncolytic virotherapy.

### 3.1. ER Stress Inducers

One approach to promote tumor cell death is by amplifying ER stress. When cells synthesize secretory proteins in amounts that exceed the processing machinery, proteins are accumulated in the ER. Because this setting is linked to cells with high protein synthesis levels such as cancer cells and virally infected cells [155,156], OV-infected tumor cells would be particularly sensitive to disruption of ER homeostasis. The protein accumulation triggers the unfolded protein response (UPR) which tries to alleviate the ER by increasing ER chaperone gene transcription, lowering protein synthesis, and, if all else fails, inducing cell death (Figure 4C) [157]. Inhibiting these adaptive UPR measures has been studied in combination with the oncolytic M1- and adenovirus using the valosin-containing protein (VCP) inhibitor Eeyarestatin I and the Golgi-specific brefeldin A-resistant guanine nucleotide exchange factor 1 (GBF-1) inhibitor golgicide A (GCA-2), respectively. These combinations resulted in the significantly enhanced anticancer efficacy of the OV treatment [158,159]. The fine balance between homeostasis and apoptotic induction by the UPRER, now requires more mechanistic knowledge of virus interactions with the UPRER and drug synergy experiments, before this field is ripe for clinical applications [160]. Indirect effects of ER stress inducers, such as thapsigargin (Tg) and ionomycin (Im), can also enhance the activity of oncolytic adenoviruses through an alteration in Ca2+ flux and protein kinase C signaling [161].

### 3.2. Analogues of DNA Building Blocks

Pyrimidine analogues, such as Gemcitabine and 5-fluorouracil, are common chemotherapeutic compounds used for treating various types of malignancies. By interfering with DNA replication these antimetabolites induce inhibition of DNA synthesis with subsequent p53 upregulation, which ultimately can lead to cell death (Figure 4C) [162]. Naturally, these cytotoxic compounds combine well with several OVs [163,164,165,166,167,168,169]. However, these antimetabolites can also induce senescence of tumor cells which can regain proliferative activity after treatment cessation [170]. Here certain OVs, like oncolytic measles virus, have been shown to contribute to eliminating these senescent cells, thereby avoiding relapse [167]. Specific pyrimidine analogues can also have immune modulating effects. These have been suggested to positively affect the antitumor immune response over the antiviral one [166].

### 3.3. Antagonizing Inhibitors of Apoptosis (IAPs)

One major barrier to effective OV therapy is virus-induced expression of type I IFN and nuclear factor kappa B (NF-κB)-responsive cytokines, which can orchestrate an antiviral state in tumors. On the other hand, the subsequently produced cytokines (TNF-α, Fas ligand (FasL), TNF-related apoptosis-inducing ligand (TRAIL), etc.) can also be exploited to induce tumor cell killing beyond the zone of initial infection, facilitated via co-treatment with a number of different pharmaceutical agents, such as SMAC-mimetic compounds (SMCs) [154,171,172] and B cell lymphoma-2 (BCL-2) homology domain 3 (BH3) mimetics [173,174]. Of note, tumor cells are often more sensitive to these chemical compounds than normal cells since NF-κB signaling is frequently constitutively activated [175], leading to elevated expression of proteins participating in cell death pathways [176].

The second mitochondria-derived activator of caspase (SMAC) is a pro-apoptotic factor released from the mitochondria during the process of cell death. Cytosolic SMAC can potentiate the activity of different caspases by inhibiting X-linked inhibitor of apoptosis protein (XIAP) and cellular inhibitors of apoptosis (cIAPs) (Figure 4B), which otherwise antagonize caspase cleavage [177]. SMAC mimetic compounds (SMCs) are small molecule mimetics of this cellular factor that can potentiate TRAIL- and TNF-α-mediated cell death (Figure 4A,B), especially in tumor cells where theses signaling pathways are aberrant [178]. Despite their potent effects on certain cell lines as a single agent due to the presence of endogenous TNF-α, SMAC mimetics are ineffective as a monotherapy in most tumor cell lines. In addition, drug resistance mechanisms include a SMC-induced upregulation of cIAP2 [179] and LRIG1 [180]. As enhancers of pro-apoptotic stimuli, however, they act as strong enhancers of the cytotoxicity of many apoptosis-inducing therapies, such as OVs [181]. This synergy has been described for several SMCs and viruses (see Table 2) and is mainly facilitated by the cytokines produced in response to OV infection. The most important cytokines involved are TRAIL [178,182,183], IL-8 [183], IL-1A [183], IL-1β [184] and TNF-α [176,185]. To improve the synergy between SMC and OVs even further, OVs have been armed with exogenous tumor cell death enhancing (TCDE) cytokines, like TNF-α [186], which also addresses toxicity issues commonly associated with their systemic delivery. In an armed OV setting, production of these cytokines is largely limited to the tumor [187].

Apart from enhanced cytotoxic effects, SMC/OV combinations can also improve the antitumor response by modulating the adaptive immune response. Exhaustion of CD8^+^ T-cells was reduced by an SMC-induced tumor macrophage M2 to M1 repolarization, an effect that could be further enhanced by PD-1 checkpoint blockade [190].

B cell lymphoma-2 (BCL-2) homology domain 3 (BH3) mimetics are antagonists that can bind with the hydrophobic Bcl-2 homology (BH) groove of Bcl-2 family proteins, thereby inhibiting these pro-survival proteins and restoring the apoptotic processes in tumor cells (Figure 4B) [192]. Several BH3 mimetics, namely GX15-070 (Obatoclax), EM20-25, BI-97D6 were shown to synergistically increase tumor cell death when combined with oncolytic vaccinia virus, VSV and AdV, respectively [173,174,193,194]. BH3 mimetics also could have a place in the cancer vaccine setting where treatment with GX15-070 (Obatoclax) increased intra-tumoral activated CD8^+^ T-cells while reducing Treg activity [193].

### 3.4. Microtubule Targeting Compounds

Taxane compounds achieve their therapeutic effect through stabilizing the spindle microtubule dynamics resulting in inhibited cell division (Figure 4D) [195]. In combination with OVs, the microtubule stabilizing agents (MSAs), docetaxel and paclitaxel, were able to sensitize a variety of tumor types to cell death following stimulation by a subset of OV infection-induced cytokines [196,197,198,199,200,201,202]. In combination with reovirus, even tumor cells not sensitive to paclitaxel alone showed a strongly enhanced cell death, which was less due to increased oncolyis but, rather, resulted from activation of cell death programs prior to viral assembly [203]. OVs, armed with pro-apoptotic cargos, could sensitized the cancer cells even further to combination treatment [204]. More out-of-the-box ideas, such as encapsulating paclitaxel and oncolytic adenovirus, together in extracellular vesicles with improved transduction and efficacy, show that there new modes of synergy still to be elucidated [205].

Another way of interfering with the tubuline network is through destabilization. Indeed, microtubule-destabilizing agents (MDAs), such as vinca alkaloids, colchicine and platinum compounds, have long been used as cancer chemotherapeutics. These compounds can also increase cell death through bystander killing after exposure to OV-induced cytokines [206,207,208]. The synergy of these types of compounds have been described in numerous animal and human settings [200,201,203,208,209,210,211,212,213]. In addition, MDAs were able to increase OV replication through a previously unappreciated role of microtubule structures in regulating type I IFN translation (Figure 1). A colchicine-induced drop in IFN and ISG expression allowed for a more robust replication of an oncolytic VSV variant with a heightened IFN sensitivity [206,214]. On the other hand, HSV-induced cisplatin retention was reported, resulting in increased DNA damage and anti-tumor immunity [215]. An additional route through which OV treatment can facilitate cell death in combination with chemotherapeutics, more specifically platinum compounds, is by downregulating myeloid cell leukemia 1 (MCL-1) (Figure 4B). MCL-1 is an anti-apoptotic member of the BCL-2 protein family that is more strongly degraded during oncolytic adenovirus infection. Its elimination in turn allows compounds like cisplatin to push tumor cells more efficiently towards cell death [216].

### 3.5. Topoisomerase Inhibitors

DNA topoisomerases are enzymes that solve topological problems associated with DNA replication, transcription, recombination, and chromatin remodeling by introducing temporary single- or double-strand breaks in the DNA [217]. Topoisomerase inhibitors are small molecules that interfere with the function of these enzymes through either intercalation or alkylation, leading to single and double stranded DNA breaks (Figure 4C). When the integrity of the genome is sufficiently compromised, apoptosis and cell death will follow, particularly in fast dividing cells, such as tumor cells, which are especially sensitive to this [218,219]. Improving the potency of these inhibitors, specifically in tumor cells, could allow lower dosing of these compounds, thereby limiting their adverse effects. This is of special importance for these therapeutics, since their use has been linked to the development of leukemia later in life [220,221]. An important mode of action of the reported synergy between OV treatment and doxorubicin is believed to be both treatments pushing the tumor cells in conflicting states of mitotic progression, resulting in higher tumor cell death than either monotherapy could achieve [222]. In addition, the effect of doxorubicin can be augmented by OV-mediated MCL-1 downregulation with co-treatment significantly increasing tumor cell death (Figure 4B,C) [223].For several cancer types, doxorubicin-treated senescent tumor cells, which are resistant to more classical methods of treatment, were efficiently killed by an oncolytic measles virus [167]. The combination of doxorubicin with an oncolytic adenovirus improved cell death in a more immunogenic fashion. This was further enhanced with additional co-treatment of the cyclophosphamide analogue ifosfamide [224]. Alternatively, the co-application of doxorubicin can also promote an increased infectivity of tumor cells by oncolytic viruses such as certain reovirus strains [225,226]. A more complex interplay has also been reported, where OV treatment induces the nuclear translocation of the cytoplasmic transcription factor cAMP response element-binding protein 3-like 1 (CREB3L1) [227], which in turn is associated with augmented doxorubicin-mediated cell death [228].

## 4. Combinations Improving the Antitumor Immune Response

Although initially envisioned to act primarily via their tumoricidal actions, over the last decade oncolytic viruses have emerged as potent immune activators and promising partners for cancer immunotherapies. The potential and promising preclinical and clinical findings of combinations of OVs with major immunotherapeutic approaches such as immune checkpoint inhibitors, T cell therapies, and cancer vaccines are beyond the scope of this small molecule themed review but are extensively discussed in recent publications [22,229,230,231,232,233]. Small molecule compounds that augment the antitumor immune response can modulate the tumor microenvironment or affect the adaptive immunity arm. The natural immune-activating characteristics renders OVs as the ideal platform to work in conjunction with small molecule immunotherapies. The TME consists of extracellular matrix (ECM), stromal and immune cells. Some of these cells such as Tregs, MDSCs and M2 macrophages drive an immunosuppressive environment by the secretion of cytokines such as IL-10 or TGF-β [234,235]. Within the TME many human tumors are infiltrated by Tregs [236], with preclinical data indicating that their depletion can enhance or restore anti-tumor immunity [237]. This makes Treg-depleting small molecules attractive candidates to counter cancer relapses caused by these immunosuppressive cells after OV treatment.

### 4.1. Cyclophosphamide (CP)

CP was extensively tested in combination with OVs, where synergy was described mostly through CPs immunosuppressive effects which allowed the OVs to replicate longer, thereby prolonging and enhancing their therapeutic efficacy [238,239,240,241]. However, CP can also play a role in improving the anti-tumor immune response elicited by initial OV treatment. Low-dose CP does not have the same immunosuppressive and toxic effects that allow increased OV replication, but does decrease the number of Tregs without compromising induction of antitumor or antiviral T-cell responses [242,243]. This selective sensitivity of Tregs to CP, comprehensively reviewed by Madondo et al. [244], works through several mechanisms. Combined, these mechanisms allow for depletion or reduced activity of Tregs, while leaving other cell populations intact [244]. This approach shows great promise, especially in combination with oncolytic virus-based cancer vaccination [245].

### 4.2. Inhibitors of VEGF and PDGF Signaling

VEGF-targeting agents such as sunitinib and cabozantinib can modulate the composition of immune cell subpopulations in the tumor and have been shown to enhance the efficacy of OV treatment. These agents, in combination with OVs, also act on several other aspects of the tumor adaptive immunity and TME, but mainly act through reducing the function of immunosuppressive cells, such as MDSCs, which in turn change cytokine levels (IL-1b, IL-6 and C-X-C motif chemokine ligand 1 (CXCL1)) and amplify the CD4^+^ and CD8^+^-mediated tumor regression [109,110,117,246]. The molecular mechanism underlying this MDSC depletion is believed to relate to inhibition of STAT3, which blocks the development of immature myeloid cells into MDSCs, and VEGFR blockade, which results in a lower capacity of MDSCs to migrate to the TME [247].

### 4.3. Transforming Growth Factor-β TGF-β Inhibition

During cancer progression, cross-talk of EGFR signaling occurs with another important signaling cascade, which is centered around the cytokine family of TGF-β [248,249]. The effects of TGF-β are very diverse and affect many signaling pathways of numerous cell types in vivo, including cancer cells [249]. Due to the interaction complexity, the effect of TGF-β evolves throughout the progression of cancer. Initially, it has a suppressing effect by triggering cell cycle arrest [250]. However, as cancer progresses, tumor cells become resistant to this response and TGF-β signaling results in epithelial–to-mesenchymal transition and increased cell migration with subsequent metastases [250,251]. TGF-β also contributes to an immunosuppressive TME [252], which impedes any anti-tumor immune response that is elicited during OV treatment [253]. Indeed, when a small-molecule inhibitor of TGF-β receptor 1 (TGF-βR1), known as A8301 [254], was combined with oncolytic HSV as treatment for murine rhabdomyosarcoma, an increased efficacy was seen due to an improved anti-tumor T cell response [255]. During non-canonical TGF-β signaling, crosstalk occurs with numerous other signaling pathways, such as PI3K, JNK and NF-κB [249]. As described above, these signaling pathways can have inhibiting effects on the replication and potency of OVs. In certain tumor settings an indirect inhibition of the pathways through TGF-β blockage could also promote OV replication. Indeed, in glioblastoma (GBM) the TGF-βRI kinase inhibitors, galunisertib [256], SB431542 and LY2109761 facilitated an increase in HSV replication through indirect inhibition of JNK-MAPK signaling [257]. Interestingly, SB431542 also inhibited oncolytic reovirus-mediated cell lysis, contrary to A8301 and galunisertib (LY2157299), indicating TGF-β signaling independent mechanisms further to be elucidated [258].

### 4.4. Topoisomerase Inhibitors

The cytotoxicity of some topoisomerase inhibitor compounds has been shown to be associated with enhanced immunogenicity of dying cells, in part due to the widespread genomic damages [259]. In addition, topoisomerase inhibitors can also improve tumor immunogenicity by upregulating antigen presentation as shown for a variety of melanoma cell lines and gliomas in response to nanomolar levels of DNA intercalating daunorubicin [260]. These immune activating characteristics could be synergistically enhanced by a combination of an oncolytic herpesvirus and adenovirus with mitoxantrone [261] and temozolomide [27,262,263].

### 4.5. Novel Compounds Targeting Adaptive Treatment Resistance of the Tumor

There are also numerous other small molecule inhibitors that counteract different aspects of immunosuppressive adaptive-mediated treatment resistance. However, these compounds have yet to be tested in combination with OVs and will therefore only be mentioned briefly, for example, inhibition of ubiquitin-specific peptidase 7 (USP7) [264,265], PI3Kdelta [266], the CBP/EP300. In addition, topoisomerase inhibitors can also improve tumor immunogenicity by upregulating antigen presentation as shown for a variety of melanoma cell lines and gliomas in response to nanomolar levels of DNA intercalating daunorubicin [260] or bromodomain [267]; all have been shown to inhibit Treg function, subsequently allowing for a more potent antitumor immune response to arise. 

### 4.6. Checkpoint Inhibitors (CPIs)

The benefits of combining antibody-based CPIs with OVs are well-known and have been comprehensively reviewed elsewhere [22,268,269,270]. Naturally, upregulation of immune checkpoints is a common result after OV treatment, leading to an increase in immune suppression and subsequent tumor relapse [32]. This can be countered by macromolecule CPIs. However, small molecule CPIs have also been developed and hold several benefits over their antibody counterparts. This upcoming class of small molecules has been extensively reviewed [271,272,273,274]. However, combinations with OVs have not yet been described for small molecule CPIs. 

### 4.7. Stimulator of Interferon Genes (STING)

The cyclic guanosine monophosphate–adenosine monophosphate (GMP-AMP) synthase (cGAS)-stimulator of the interferon genes (STING) signaling pathway has recently been described as playing an important role, not only in the innate response to infection [275,276,277,278], but also in cancer immune surveillance. STING activation initiates a type I interferon (IFN)-driven pro-inflammatory program that stimulates basic leucine zipper transcriptional factor ATF-like 3 (BATF3)-dependent dendritic cell (DC) cross-presentation and promotes CD8^+^ T cell-mediated anti-tumor immune responses [279,280,281,282]. STING agonists have thus emerged as a class of promising new therapeutics that may enhance tumor immunogenicity and several candidates are being evaluated in pre-clinical and clinical contexts [283,284,285]. However, STING deficiency is common in several cancer entities due to the anti-tumorigenic and immune-activating role of STING signaling [286,287,288] and data suggest that, consequently, oncolytic viruses benefit from STING loss due to a decreased antiviral IFN response [287,288]. Several OVs also encode gene products that interfere with the cGAS–STING signaling pathway [289,290]. These considerations make a potential combination of OV with STING agonists at first look counterintuitive. However, STING deficiency or dysfunction has been associated with an exclusion of lymphoid cells from the TME [279] and, while viral replication may be enhanced in STING loss tumors, an optimal induction of an adaptive anti-tumor immune response could be hindered. Indeed, OVs that induce an IFN response via cGAS-STING signaling may have an advantage due to the involvement of this pathway in the bridging of innate and adaptive immunity [291]. Hence, the combination of small molecule STING agonists with certain oncolytic viruses may represent an interesting novel approach to enhance anti-tumor immune responses in OV therapy, although careful assessment of the co-treatment regimen to balance the antiviral and antitumoral effects of STING will be paramount.

## 5. Safety Considerations

To date, clinical experience with virotherapy-enhancing combinations is limited and our current understanding on the synergism of select combinations has been based on extensive preclinical studies. Twenty years of clinical testing of OV’s in monotherapy settings have underlined their excellent safety profile with grade 1 and 2 being the most commonly reported adverse events [5]. To what extent some small molecule combinations may compromise such a safety profile or adversely affect the overall therapeutic efficacy of oncolytic viruses is currently, in large part, subject to conjecture and should therefore be carefully addressed in pre-clinical settings. For example, dimethyl fumarate potentiates replication and oncolysis induced by VSVΔM51 [66], but lowers leukocyte counts and can result in reactivation of JC virus, leading to multifocal leukoencephalopathy (PML). Some HDIs have also been shown to reactivate latent HIV [292], EBV and HSV-1 [293]. The risk that such compounds may reactivate a second virus, with that virus’ interactions with the initial oncolytic virus being unknown, should not be underestimated. The specific inhibition profiles of the particular small molecule, as well as the OV in question, will also determine the outcome of an OV/drug combination. While enhancing OV replication, inhibition of certain HDACs (HDAC 2, 6, 11) may enhance Treg function [294], so choosing a drug with a favorable profile, selection of patients with low tumor Treg counts or careful scheduling of the drug and OV may enhance the final anti-tumor synergy. In addition, some virotherapy-enhancing combinations may also potentially enhance the safety profile. For example, ruxolitinib has long been proposed to enhance activity of numerous OVs due to countering the antiviral JAK/STAT signaling and no toxicities have been reported in different preclinical studies [44,49]. However, its combination with an interferon-armed VSV-hIFN-NIS in two current clinical trials (see Table 3) may also act to offset potential toxicities caused by excessive production of the interferon transgene in particularly permissive tumors.

## 6. Conclusions

While our understanding of how to capture the full potential of oncolytic virotherapy continues to evolve, it appears clear that release of tumor associated antigens and activation of the immune system is crucial for these anti-oncolytic agents. Consequently, combinations of oncolytic viruses with immune checkpoint inhibitors are dominating the current clinical trial landscape [295,296]. However, combinations with select small molecule compounds can address some of the limitations of the oncolytic core features and improve oncolysis, intra-tumoral spread, immunogenicity of tumor cell killing, as well as improving antigen processing and the regulation of immune cell populations. Such combinations have now also entered clinical testing [18] (for currently active trials, see Table 3).

In conclusion, there are many potent compounds available to counter most immunosuppressive mechanisms a tumor can display. The big challenge will be to develop methods to efficiently and affordably determine which combination to use when, and for which patients.

## Figures and Tables

**Figure 1 cancers-13-03386-f001:**
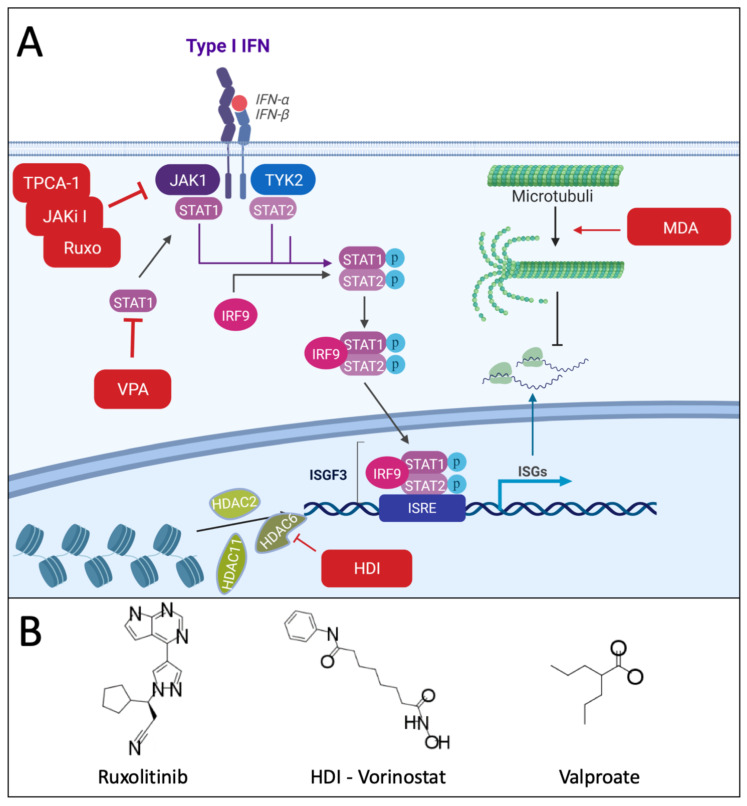
JAK/STAT signaling inhibition for the improvement of OV efficacy. (**A**) IFN binding with its receptor can activate JAK1 and TYK2. This in turn facilitates the phosphorylation of the docking sites of STAT1 and STAT2. Following phosphorylation, both STATs associate with IRF6 to form the transcriptional regulation ISG3. ISG3 trans-locates to the nucleus where it mediates the transcription of ISG mRNAs. The appropriate DNA strains are made accessible for ISGF3 by different histone deacetylases. These mRNAs are in turn transported over microtubules in order to be translated. Targeting these pathways by means of different small molecule inhibitors (red annotated squares) allows OV replication to proceed for longer, resulting in increased viral spread and potentially efficacy. See the main text for more details. Created with biorender.com. (**B**) Selected chemical structures of compounds depicted in panel A. All structures throughout were drawn using MarvinSketch (ChemAxon) from publicly available information. Abbreviations: JAK, Janus kinase; STAT, signal transducers and activators of transcription; IRF9, Interferon regulatory factor 9; ISGF3, Interferon-stimulated gene factor 3; HDAC, histone deacetylase; ISRE, Interferon-sensitive response element; MDA, microtubule destabilizing agent; VPA, Valproate.

**Figure 2 cancers-13-03386-f002:**
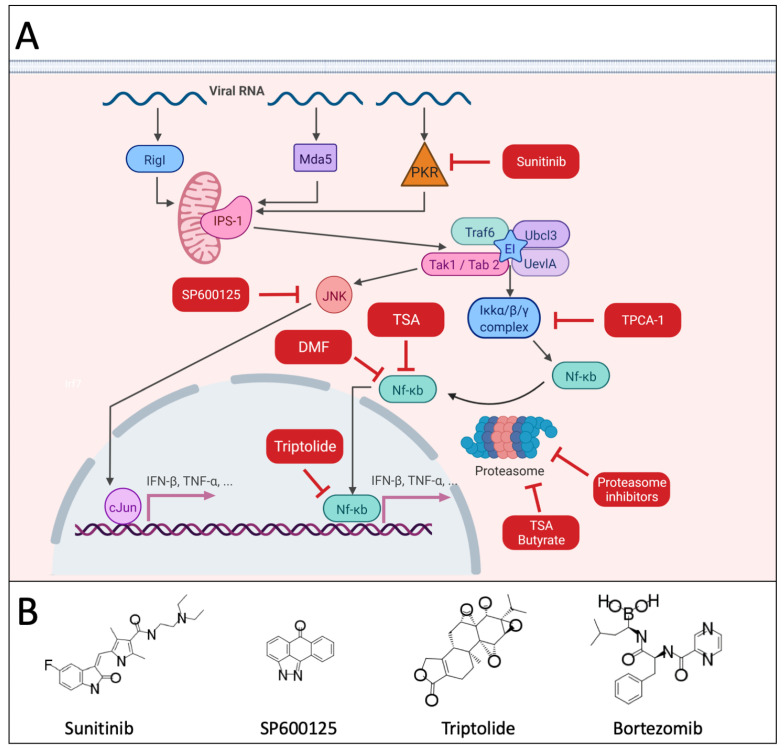
Compound classes that inhibit of NF-kB signaling and synergize with OV treatment. (**A**) Virus replication results in the production of cytosolic DNA and single- and double-stranded RNA. This triggers multiple signaling cascades, including the recruitment of RIG-I and Mda5 to the adaptor IPS-1 on the membrane of the mitochondria. This in turn leads to kinase activation through TRAF family members. More specifically, this activates the IKK complex, which phosphorylates IκB proteins. Phosphorylation of IκB leads to its ubiquitination and proteasomal degradation, freeing NF-κB complexes for transcription induction. TRAF6 signaling also leads to JNK activation. Activated JNK trans-locates to the nucleus and activates c-Jun and other target transcription factors. These transcription factors, such as cJun and NF-κB lead to the transcription of numerous proteins involved in innate immunity and cells death, including IFN-β. Interfering with the different steps of signaling pathways using different classes of compounds (red annotated red squares) have resulted in increased viral replication and subsequent efficacy. See the main text for more details. Created with biorender.com. (**B**) Selected chemical structures of compounds depicted in panel A. All structures throughout were drawn using MarvinSketch (ChemAxon) from publicly available information. Abbreviations: TRAF, TNF Receptor Associated Factor; JNK, c-Jun N-terminal kinase; Atf2, Activating transcription factor 2; IPS-1, interferon-β promoter stimulator 1; TSA, Trichostatin A; DMF, dimethyl fumarate; RigI, retinoic acid-inducible gene I; mda5, melanoma differentiation-associated protein 5; PRK, protein kinase R; Ubcl3, ubiquitin-conjugating enzyme 13; ubiquitin-conjugating enzyme E2 variant 1; Tak1, transforming growth factor-6-activated kinase 1; IKK, IκB kinase β.

**Figure 3 cancers-13-03386-f003:**
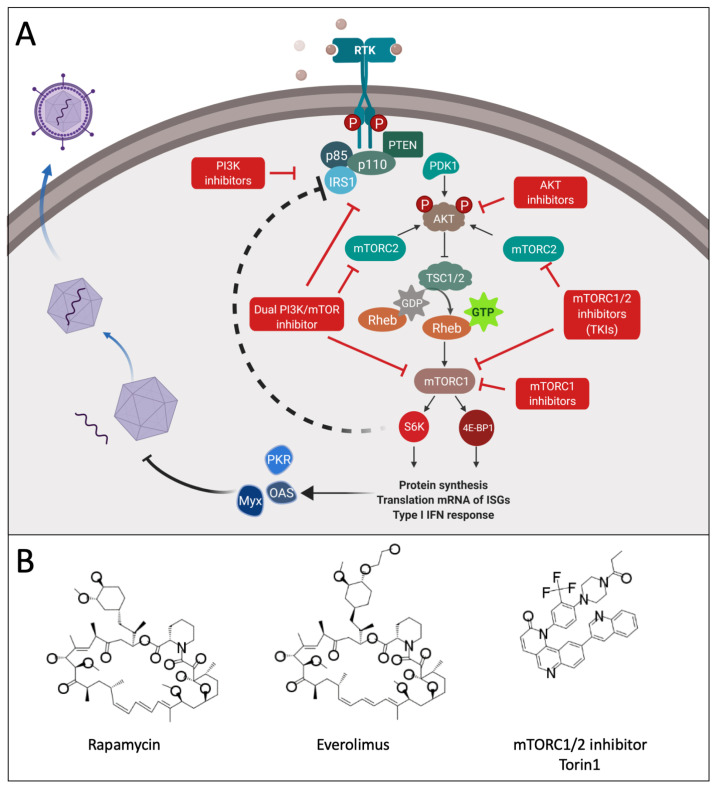
Overview of the PI3K(p85/p110)/AKT/mTOR pathway and small molecule compounds that target this pathway in synergy with OV therapy. (**A**) Activating (PI3K, AKT, PDK1, mTORC1 and mTORC2) and inhibiting proteins (PTEN, TSC1/2) of the signaling pathway are shown. PI3K consists of catalytic subunit p110 and the regulatory subunit p85. PI3K phosphorylates phosphatidylinositol bisphosphate, which in turn activates PDK1 and AKT. PTEN negatively regulates the activation of AKT, which can inhibit TSC1/2, a negative regulator of mTOR. Active mTOR phosphorylates S6K1 and 4EBP1 leading to increased translation and synthesis of, among others, ISGs [73]. Targeting this process by means of different small molecule inhibitors (red annotated squares) allows OV replication to proceed for longer, resulting in increased viral spread and efficacy. See the main text for more details. Created with biorender.com. (**B**) Selected chemical structures of compounds depicted in panel A. All structures throughout were drawn using MarvinSketch (ChemAxon) from publicly available information. Abbreviations: RTK, receptor tyrosine kinase; PDK1, phosphoinositide-dependent kinase 1; IRS1, insulin receptor substrate 1; PTEN, phosphatase and tensin homologue; mTOR, mammalian target of rapamycin. PKR, protein kinase R; Myx, GTP-binding protein MX; AOS, oligoadenylate synthetase; S6K, S6 kinase; 4E-BP1, Eukaryotic translation initiation factor 4E-binding protein 1; Rheb, Ras homolog enriched in brain; IRS1, insulin receptor substrate 1.

**Figure 4 cancers-13-03386-f004:**
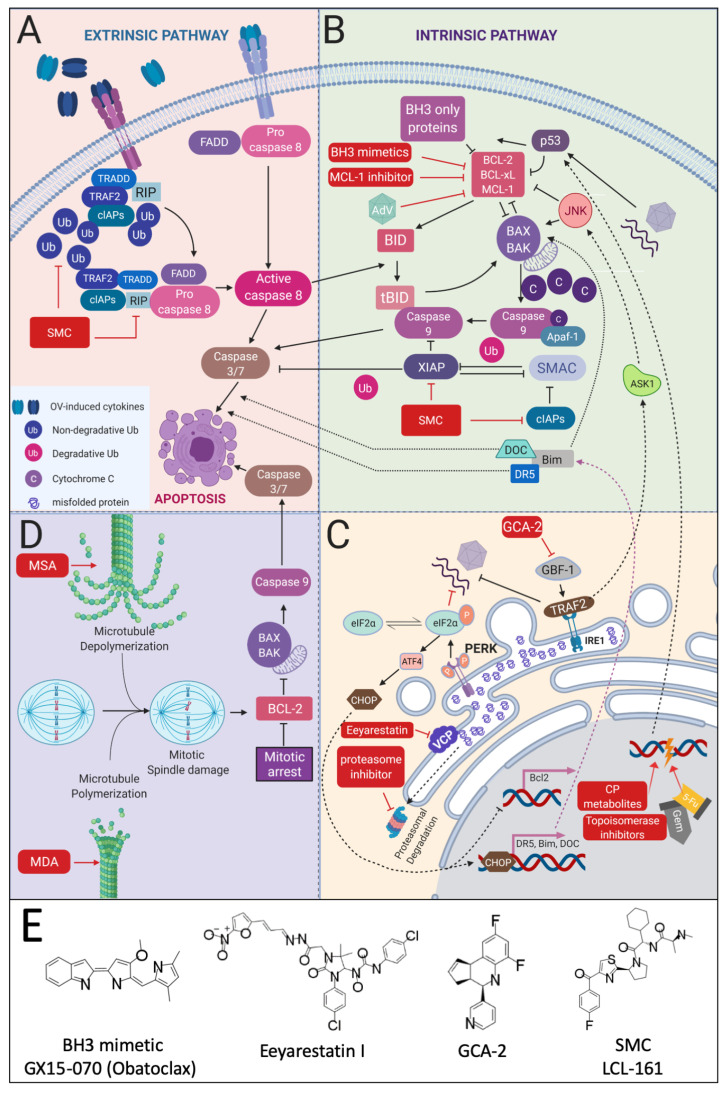
Increasing bystander killing of tumor cells by small molecules after OV treatment. (**A**) Cytokines produced in response to OV treatment of the tumor can activate the extrinsic pathway for apoptosis through binding with death receptors such as Fas and TNF-α receptor. Oligomerization of these receptors in turn facilitates the recruitment of adaptor proteins, for example, binding of Fas ligand with Fas recruits caspase-8 through the adaptor protein FADD. Cleaved caspase-8 can directly activate caspase-3 and result in cell death. (**B**) Additionally, cleaved caspase-8 connects to the pathways of intrinsic apoptosis. This occurs when it cleaves Bid. Truncated Bid subsequently trans-locates to the mitochondria where it induces cytochrome release leading to activation of caspase-9 and caspase-3. This cytochrome c release is facilitated by the oligomerization of the pro-apoptotic Bax and Bak proteins at the outer mitochondria membrane. This process stands under the control of several proteins including Bcl-2, Bcl-xL and MCL-1. These pro-survival proteins in turn are inhibited by “BH3 only” proteins. (**C**) Intrinsic apoptosis can also be additionally stimulated through compounds that induce DNA damage, since this leads to p53 upregulation, resulting in indirectly Bax/Bak activation. ER stress signaling, caused by the accumulation of misfolded protein in the ER, can also facilitate this effect through ASK1 with the activation and subsequent translocation of JNK to the mitochondrial membrane. In addition, ER stress can also promote cell death through the activation of MAPK-mediated activation of eIF2α and ATF4 leading to the nuclear translocation of CHOP where it promotes transcription of pro-apoptotic genes. Apart from promoting cell death, eIF2α and TRAF2 also attenuates protein translation when misfolded protein accumulate in the ER. Since this is often the case during OV replication, the inhibition of these mechanisms can improve the efficacy of OV treatment. (**D**) Also the stabilizing or destabilizing of microtubules can trigger apoptosis. More specially, when cells are arrested G2/M phase, this can lead to the activation of intrinsic apoptosis. Targeting these pathways can improve oncolysis, tumor immunogenicity and viral replication depending on what aspect of cell death is targeted. Small molecule compounds targeting different stages of this process are presented by red annotated squares. See the main text for more information. Created with biorender.com. (**E**) Selected chemical structures of compounds depicted in panels A-D. All structures throughout were drawn using MarvinSketch (ChemAxon) from publicly available information. Abbreviations: TRADD, TNFR1-associated death domain protein; TRAF2, TNF receptor-associated factor 2; cIAP, cellular inhibitor of apoptosis; RIP, receptor interacting protein; FADD, fas-associated death domain; BH3, BCL-2 homology domain 3; SMC, Second mitochondria-derived activator of caspase mimetic compounds; Ub, ubiquitin; MCL-1; myeloid cell leukemia 1; XIAP, X-linked inhibitor of apoptosis protein; BID, BH3 interacting-domain death agonist; tBID, truncated Bid; AdV, adenovirus; JNK, c-Jun NH2-terminal kinase; BCL-xL, B-cell lymphoma, extra-large; BCL-2, B-cell lymphoma 2; BAX, BCL2 associated X; BAK, Bcl-2 homologous antagonist killer; Apaf-1, apoptotic protease activating factor-1; ASK1, Apoptosis signal-regulating kinase 1; CHOP, CCAAT-enhancer-binding protein homologous protein; DOX, downstream of CHOP; DR5, death receptor 5 (DR5); MDA, microtubule-destabilizing agents; MSA, microtubule-stabilizing agent; ATF4, Activating transcription factor 4; PERK, PRKR-like endoplasmic reticulum kinase; IRE1, inositol-requiring enzyme; CP, cyclophosphamide; Gem, gemcitabine; 5-Fu, fluorouracil; GBF-1, Golgi-specific brefeldin A-resistant guanine nucleotide exchange factor 1; GCA-2, GBF-1 inhibitor golgicide A; P, phosphorylated; VCP, valosin-containing protein; eIF2α, eukaryotic translation initiation factor 2α.

**Table 1 cancers-13-03386-t001:** Synergy of HDIs and OVs.

HDI	OV	Tumor	References
entinostat	VSV	B16-F10, CT26, L363(MM), HT29, M14, PC3, SW620, 4T1	[17,129,130,140]
vorinostat	VSV	B16-F10	[130,141]
trichostatin	HSV, vaccinia	SAS, Ca9-22, HSC, HCT116, B26-F10, U87, SW480, HeLa	[123,142,143]
valproate	HSV, H1	U87, AGS1, U251, Gli36, HeLa	[132,133,134,144]
Scriptaid & LBH589	Adenovirus	Glioblastoma	[145]

**Table 2 cancers-13-03386-t002:** Selected SMC/OV combinations.

SMC	OV	Tumor Model	References
LCL-161	VSV, M1	EMT-6, CT26, MOC-11, SNB75, SG539, BTIC, HCT-116, Kym-1, M-3	[154,183,186,188,189,190,191]
Birinapant	M1	HCT-116, Huh-7	[183,191]

**Table 3 cancers-13-03386-t003:** Currently active * clinical trials with oncolytic virus and small molecule compound combinations.

Virus Family	Oncolytic Virus Design	Small Molecule Compound	Indication	Phase/Status	CinicalTrials.gov Reference
HSV	rQNestin34.5v.2 HSV-1 with viral gene ICP34.5 under glioma specific nestin promoter control	Cyclophosphamide	Glioma	I recruiting	NCT03152318
	TBI-1401(HF10) naturally attenuated HSV-1	Gemcitabine + nab-pactitaxel	Pancreatic cancer	I not recruiting	NCT03252808
AdV	ONCOS-102 Ad5/3-24 expressing a GM-CSF transgene	Cyclophosphamide	Melanoma	I not recruiting	NCT03003676
	ONCOS-102 Ad5/3-24 expressing a GM-CSF transgene	Cyclophosphamide	Mesothelioma	II not recruiting	NCT02879669
	LOAd703 AdV5/35 expressing TMZ-CD40L and 4-1BBL transgenes	Gemcitabine + nab-pactitaxel	Pancreatic cancer	I/IIa recruiting	NCT02705196
RV	Pelareorep Unmodified human reovirus typ 3 (Dearing strain)	Paclitaxel	Breast cancer	II recruiting	NCT04215146
	Pelareorep Unmodified human reovirus typ 3 (Dearing strain)	Carfilzomib	Multiple myeloma	I recruiting	NCT03605719
VV	JX-594 (Pexa-Vec) Wyeth strain VV expressing a GM-CSF transgene	Cyclophosphamide	Sarcoma, breast cancer	II recruiting	NCT02630368
VSV	VSV-hIFN-NIS VSV expressing an interferon and a sodium iodide symporter transgene	Ruxolitinib	Multiple myeloma, AML, T-cell lymphoma	I recruiting	NCT03017820
	VSV-hIFN-NIS VSV expressing an interferon and a sodium iodide symporter transgene	Ruxolitinib	Endometrial cancer	I recruiting	NCT03120624

AdV, adenovirus; GM-CSF, granulocyte-macrophage colony-stimulating factor; hIFN, human interferon; HSV-1, herpes simplex virus type 1; ICP, infected cell protein; TMZ-CD40L, trimerized membrane-bound CD40 ligand; VSV, vesicular stomatitis virus; VV, vaccinia virus. * clinicaltrials.org accessed on 23 June 2021; search term “oncolytic”; filters “recruiting” and “active, not recruiting”.

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
