# Peer review of "Combining Oncolytic Viruses and Small Molecule Therapeutics: Mutual Benefits"

_cancers, 2021, doi:10.3390/cancers13143386_

Round 1

Reviewer 1 Report

A complete and well written review of potential  OV and small molecules. This reviewer would have found a table of completed and ongoing clinical trials helpful to give the reader an appreciation of  the clinical activity in this space.

A more exhaustive review of potential toxicities with these combinations should be included.

Although the references are quite exhaustive very few are more recent than than 2018 are included giving some pause around how up to date this review really is.

Author Response

Referee #1:

A complete and well written review of potential OV and small molecules. This reviewer would have found a table of completed and ongoing clinical trials helpful to give the reader an appreciation of the clinical activity in this space.

We agree with the referee and added a new TABLE 3 to the manuscript listing and detailing all currently active clinical trials on small molecule – OV combinations (clinicaltrials.org accessed in June ‘21). We also cite a recent publication that also includes a list of completed trials. We also refer to a number of review articles that focus on OV – checkpoint inhibitor combinations, which are much more common but have not been the focus of this review.

In chapter 6 Conclusion:

Consequently, combinations of oncolytic viruses with immune checkpoint inhibitors are dominating the current clinical trial landscape (Chiu et al. 2020; Hwang, Hong, and Yun 2020). However, combinations with select small molecule compounds can address some of the limitations of the oncolytic core features and improve oncolysis, intratumoral spread, immunogenicity of tumor cell killing, as well as improve antigen processing and regulation of immune cell populations. Such combinations have now also entered clinical testing (Phan et al. 2018) (for currently active trials see Table 3).

A more exhaustive review of potential toxicities with these combinations should be included.

We appreciate this important suggestion and have added a new chapter “5. Safety considerations”. However, as we discuss in this paragraph, some interaction may be human specific and cannot be adequately addressed preclinically (such as the risk of reactivating unrelated human viruses). Also, current preclinical studies have so far focused mostly on mode of action and efficacy studies and preclinical studies with a focus on safety have not been published. We emphasize that such studies are pertinent before clinical testing can commence. We also discuss that some small molecule combinations might in fact also hold the potential to INCREASE the safety of certain oncolytic viruses, exemplified by the current clinical evaluation of ruxolitinib in combination with VSV-hIFN-NIS.

New chapter 5:

5          Safety considerations

To date, clinical experience with virotherapy-enhancing combinations is limited and our current understanding on the synergism of select combinations has been based on extensive preclinical studies. 20 years of clinical testing of OV’s in monotherapy settings have underlined their excellent safety profile with grade 1 and 2 being the most commonly reported adverse events (Macedo et al. 2020). To what extend some small molecule combinations may compromise such safety profile or adversely affect the overall therapeutic efficacy of oncolytic viruses is currently in large part subject of conjecture and should therefore be carefully addressed in pre-clinical settings. For example, dimethyl fumarate potentiates replication and oncolysis induced by VSVΔM51 (Selman et al. 2018), but lowers leukocyte counts and can result in reactivation of JC virus, leading to multifocal leukoencephalopathy (PML). Some HDIs have also been shown to reactivate latent HIV (Zaikos et al. 2018), EBV and HSV-1 (Nehme, Pasquereau, and Herbein 2019). The risk that such compounds may reactivate a second virus - with that virus’ interactions with the initial oncolytic virus unknown - should not be underestimated. The specific inhibition profiles of the particular small molecule as well as the OV in question will also determine the outcome of an OV/drug combination. While enhancing OV replication, inhibition of certain HDACs (HDAC 2, 6, 11) may enhance Treg function (L. Wang et al. 2018), so choosing a drug with a favorable profile, selection of patients with low tumor Treg counts or careful scheduling of the drug and OV may enhance the final anti-tumor synergy. In addition, some virotherapy-enhancing combinations may also potentially enhance the safety profile. For example, ruxolitinib has long been proposed to enhance activity of numerous OVs due to countering the antiviral JAK/STAT signaling and no toxicities have been reported in different preclinical studies (Patel et al. 2019; Dold et al. 2016). However, its combination with an interferon-armed VSV-hIFN-NIS in two current clinical trials (see Table 3) may also act to offset potential toxicities caused by excessive production of the interferon transgene in particularly permissive tumors.

Although the references are quite exhaustive very few are more recent than than 2018 are included giving some pause around how up to date this review really is.

We appreciate this comment and have updated our reference list accordingly. In total (updated refs and refs for additional content) 46 new references have been added. For better tracing, they have been marked with yellow highlights in the revised manuscript version.  

Reviewer 2 Report

The authors provide a comprehensive review of multiple small molecule compounds and how they interact with oncolytic viruses.  I have the following questions/comments.

1.  The manuscript would benefit from a brief general discussion or overview of the different types of oncolytic viruses (highlighting similarities/differences) mentioned in this review to provide background for readers. 

2.   Along those same lines, is there one type of oncolytic virus backbone (HSV, VSV, vaccinia, Ad, etc) that is most applicable to combination with these small molecule compounds?

3.  There are multiple instances of abbreviations being used in the body of the text without being properly defined (e.g.  PKR, ER).   These should be addressed to enhance readability of the manuscript.

4.  There are also instances of concepts/statements that could benefit from expansion to add additional clarity.  For example, line 83 "some tumors may even display an upregulated antiviral state leading to primary resistance."  Are there certain types of solid tumors (soft tissue cancers, GI malignancies, etc.) that this type of combination therapy (OV + small molecule compounds) would be most efficacious for?  Similarly, in lines 155-56, what types of OVs have enhanced replication from the inhibition of NFkB signaling?

Author Response

Referee #2:

The authors provide a comprehensive review of multiple small molecule compounds and how they interact with oncolytic viruses.  I have the following questions/comments.

  1. The manuscript would benefit from a brief general discussion or overview of the different types of oncolytic viruses (highlighting similarities/differences) mentioned in this review to provide background for readers.

We appreciate this comment and admit that due to the nature of this special edition of “Cancers” on oncolytic viruses we initially opted to minimize the OV background introduction. However, we agree that for the general readership such introduction would be beneficial. We have added a brief introduction of each of the viruses discussed in our review article. We also briefly mention the diversity of OVs and the factors that can be applied for categorization. We focus this introduction on the separation of RNA vs DNA viruses as well as viruses with strong dependence on antiviral defects in tumors.

In chapter 1 Introduction:

In general, naturally occurring or genetically engineered virotherapy candidate viruses share the core features of tumor-preferential infection, replication, and lysis. Beyond that, they display the diversity of viruses on multiple levels: human pathogen-derived versus animal viruses, DNA versus RNA genome, enveloped versus non-enveloped, nuclear versus cytosolic replication cycle etc. (Maroun et al. 2017). Herpes simplex virus (HSV) and adenovirus (AdV) are human pathogenic DNA viruses that have been developed for 3 decades as oncolytic agents with a plethora of modified variants being tested in preclinical and clinical settings. This resulted in the first regulatory approvals of H101, a genetically engineered adenovirus, in 2005 in China and talimogene laherparepvec (T-VEC), a recombinant attenuated HSV-1 with a transgene encoding for granulocyte-macrophage colony-stimulating factor (GM-CSF), in 2015 in the USA and Europe (Macedo et al. 2020). Development of oncolytic HSV and AdV variants has continued though with a strong focus on next generation “armed” OVs expressing a multitude of immune modulatory transgenes. Another clinically advanced oncolytic platform is based on vaccinia virus (VV), a large DNA virus encoding about 200 genes with exclusive cytosolic replication cycle. Its ability to accommodate up to 40kb of transgene DNA make VV a prime platform for arming with immune modulatory cargo genes (Guo et al. 2019). A related member of the poxvirus family, myxoma virus, has also extensively been explored as an oncolytic agent in pre-clinical settings (Rahman and McFadden 2020). H1, a small rat parvovirus completes the list of the major DNA-based oncolytic agents. It’s natural onco preference is in large part based on a dependency on proliferating cells and signaling pathway aberrations (Bretscher and Marchini 2019). Reovirus, a natural occurring human virus with double stranded RNA genome, is usually not associated with disease in adults and its oncotropism was originally thought to be linked to RAS transformation in cancer cells, although recent data suggest a more multifactorial relationship (Müller et al. 2020). The Edmonston vaccine strain of measles virus, a negative strand RNA paramyxovirus, displays a certain natural oncoselectivity in part due to frequent overexpression of its receptor, CD46, in a range of different cancer types (Pidelaserra-Martí and Engeland 2020). Newcastle disease virus, an avian paramyxovirus without causing known human disease, harbors a natural oncoselectivity due to interaction with anti-apoptotic proteins and its dependence on a defective antiviral make-up frequently observed in cancer cells (Tayeb, Zakay-Rones, and Panet 2015). Vesicular stomatitis virus (VSV), a negative strand RNA virus of the rhabdoviridae family, causes mild disease in lifestock with clinical symptoms rarely reported in human. Its ubiquitous receptor entry translates to a pantropism for a very broad range of tumor types, but also holds the potential for some neurotoxicty once it can access the brain. Consequently, VSV development was long driven by attenuation strategies (Felt and Grdzelishvili 2017). As with several other RNA viruses, the primary mode of oncoselectivity is based on reduced antiviral defense mechanisms in certain tumors (Stojdl et al. 2000). In recent years, a large number of VSV variants armed with immunomodulatory transgenes has been tested in preclinical settings and in early phase clinical testing (Melzer, Lopez-Martinez, and Altomonte 2017).

  1. Along those same lines, is there one type of oncolytic virus backbone (HSV, VSV, vaccinia, Ad, etc) that is most applicable to combination with these small molecule compounds?

This is not the case. While there are indeed some OVs that are expected to benefit from compounds interfering with the innate antiviral response, these same OVs may not benefit from compounds interfering with DNA replication for instance.

For clarification, we added some comments to highlight which viruses may benefit from certain compound combinations:

In chapter 1 Introduction :

With few exceptions, most OVs are rather sensitive to innate antiviral control. This increases their safety aspect towards normal cells while it lets them take advantage of impaired innate immune signaling in tumors (Stojdl et al. 2000). These OVs are therefore also considerably better suited to be combined with small molecules that counter innate antiviral immunity.

In chapter 2, sub chapter “Inhibition of NF-kB signaling”

This would be especially advantageous for OVs, like VSV and NDV, that rely on defective innate immunity for their oncoselectivity (Wollmann, Ozduman, and van den Pol 2012).

In chapter 3, sub chapter “Microtubule targeting compounds”

OVs, armed with pro-apoptotic cargos, could sensitized the cancer cells even further to combination treatment (Lal and Rajala 2019).

In chapter 4, sub chapter “Cyclophosphamide (CP)”

This approach shows great promise, especially in combination with oncolytic virus-based cancer vaccination (Pol et al. 2020).

  1. There are multiple instances of abbreviations being used in the body of the text without being properly defined (e.g. PKR, ER).   These should be addressed to enhance readability of the manuscript.

We appreciate this careful observation and have carefully proofread the manuscript and added definitions for abbreviations.

  1. There are also instances of concepts/statements that could benefit from expansion to add additional clarity. For example, line 83 "some tumors may even display an upregulated antiviral state leading to primary resistance."  Are there certain types of solid tumors (soft tissue cancers, GI malignancies, etc.) that this type of combination therapy (OV + small molecule compounds) would be most efficacious for?  Similarly, in lines 155-56, what types of OVs have enhanced replication from the inhibition of NFkB signaling?

We appreciate this suggestion to clarify these two sections.

For the first part, to our knowledge, no systemic assessment has been undertaken that allows a general statement whether there are certain tumor types more frequently displaying an upregulated antiviral state. So far, the studies on this questions have been limited to certain tumor types. We added the following clarification:

Chapter 2      Combinations affecting viral propagation in tumor cells:

Some tumors, such as pancreas cancer, may even display an upregulated antiviral state leading to primary resistance (Hastie et al. 2016). A constitutive interferon pathway activation was also described as a main determinant for oncolytic measles virus activity in human glioblastoma specimen (Kurokawa et al. 2018). On the other hand, tumors induced by oncoviruses, such as HPV-associated cervical or head and neck cancers tend to frequently display strongly impaired antiviral innate responses (Vähä-Koskela and Hinkkanen 2014). However, in light of missing systematic assessments of a large range of tumor types, general conclusions as to what cancer types are more antivirally active and which are not, remain to be drawn.

For the second part, we added the following clarification:

This would be especially advantageous for OVs, like VSV and NDV, that rely on defective innate immunity for their oncoselectivity (Wollmann, Ozduman, and van den Pol 2012).

Reviewer 3 Report

Authors described a new review of oncolytic viruses and small molecule therapeutics. The descriptions are very important and contents are very useful. However, authors should deal with a major point and a minor point. 

1. The authors should clarify the structures of all small molecules so that a wide range of readers will be interested. The authors had better show such structures in Figures 1B, 2B, 3B and 4E.

2. In line 426, what is "Error! Reference cource not found" ?

Author Response

Referee #3:

Authors described a new review of oncolytic viruses and small molecule therapeutics. The descriptions are very important and contents are very useful. However, authors should deal with a major point and a minor point.

  1. The authors should clarify the structures of all small molecules so that a wide range of readers will be interested. The authors had better show such structures in Figures 1B, 2B, 3B and 4E.

We followed this reviewer’s suggestion and added structures of the pertinent small molecules to each of the 4 figures and as suggested display them as a respective “B” part underneath each figure.

  1. In line 426, what is "Error! Reference cource not found" ?

We appreciate this catch. There may have been an issue with our reference program. The revised version should have fixed this.

Round 2

Reviewer 2 Report

Thoughtful comments and revisions to reviewers' concerns.